

# Multi-seasonal measurements of the ground-level atmospheric ice-nucleating particle abundance on the North Slope of Alaska

Aidan D. Pantoya[1], Stephanie R. Simonsen[2], Elisabeth Andrews[3], Ross Burgener[4], Christopher J. Cox[5], Gijs de Boer[6], Bryan D. Thomas[4], and Naruki Hiranuma[2]

[1]West Texas A&M University, College of Engineering, Canyon, 79016, USA

[2]West Texas A&M University, Department of Life, Earth, and Environmental Sciences, Canyon, 79016, USA

[3]Cooperative Institute for Research in Environmental Sciences, University of Colorado Boulder, Boulder, 80309, USA

[4]Global Monitoring Laboratory, National Oceanic and Atmospheric Administration (NOAA), Barrow, 99723, USA

[5]Physical Sciences Laboratory, National Oceanic and Atmospheric Administration (NOAA), Boulder, 80305, USA

[6]Environmental and Climate Sciences Department, Brookhaven National Laboratory, Upton, 11973, USA

*Correspondence to*: Naruki Hiranuma (nhiranuma@wtamu.edu)

## ABSTRACT

Atmospheric ice-nucleating particles (INPs) are an important subset of aerosol particles that are responsible for the heterogeneous formation of ice crystals. INPs modulate arctic cloud phase

(liquid vs. ice), resulting in implications for radiative feedbacks. The number of arctic INP studies investigating specific INP episodes or sources has recently increased. However, existing studies are based on short-duration field data and long-term datasets are lacking. Continuous, long-term measurements are key to determining the abundance and variability of ambient arctic INPs and for constraining aerosol-cloud interactions, for example, to verify and/or improve simulations of

mixed-phase clouds. Here, we present the first long-duration INP dataset from the Arctic: two years of immersion mode INP concentrations ($n_{INP}$) measured continuously at the National Oceanic and Atmospheric Administration's Barrow Atmospheric Baseline Observatory on the North Slope of Alaska. A portable ice nucleation experiment chamber (PINE-03), which simulates adiabatic expansion cooling, was used to directly measure the ground-level INP abundance with

an approximately 12-minute time resolution from October 2021 to December 2023. We document PINE-03 $n_{INP}$ measurements over a wide range of heterogeneous freezing temperatures from −16 to −31 °C from which we introduce new season-specific parameterizations suitable for modeling mixed-phase clouds. Collocated aerosol and meteorological data were analyzed to assess the correlation between ambient $n_{INP}$, air mass origin region, and meteorological variability. Our

findings suggest (1) very high freezing efficiency of INPs across the measured temperatures ($\approx 2 \times 10^8 - 10^{10}$ m$^{-2}$ for from −16 to −31 °C), which is a factor of $10 - 1000$ times greater efficiency





as compared to that found in the previous mid-latitude INP measurements in autumn using the same instrument; (2) surprisingly high $n_{INP}$ for the examined temperatures throughout the year that were not measured by PINE-03 at other sites; and (3) high $n_{INP}$ in spring, possibly related to arctic haze episodes.



## 1. INTRODUCTION

Ice formation in the atmosphere is facilitated by ice-nucleating particles (INPs) through heterogeneous freezing (Hoose and Möhler, 2012) by reducing the activation energy required to induce the release of latent heat, thereby triggering spontaneous ice growth (Vali et al., 2015).

Below $\approx -35\,°C$, freezing of supercooled water droplets can take place homogeneously (Koop and Murray, 2016). At warmer sub-zero temperatures, several heterogeneous freezing mechanisms are important, including immersion freezing, which is a dominant ice formation pathway in mixed-phase clouds (hereafter referred to as MPCs) (Hande and Hoose, 2017; Westbrook and Illingworth, 2011).

In the Arctic, MPCs are ubiquitous, dominating features of the low cloud fraction (Morrison et al., 2012) and radiative balance (e.g., Shupe and Intrieri, 2004). They are observed in a variety of conditions and in all seasons (e.g., Shupe et al. 2010; 2011; 2013). INPs can act as cloud-destroying agents in MPCs. For example, model sensitivity studies indicate that MPC lifetime is strongly sensitive to INP concentration ($n_{INP}$) (Solomon et al., 2018) despite the fact

that $n_{INP}$ is several orders of magnitude smaller than concentrations of cloud condensation nuclei (CCN) (Lee et al., 2023; Mamouri and Ansmann, 2016). The forcing and feedback mechanisms associated with aerosols and clouds remain uncertain (Kanji et al., 2017; Solomon et al., 2009). Murray et al. (2021) postulate that, in the Arctic, MPCs could decrease due to positive feedback with atmospheric INPs, supported by reduced snow and ice coverage enhancing INP emissions

from exposed terrestrial surfaces, or even thermokarst landforms (Barry et al., 2023).

Arctic INPs have been reported in several past studies, in particular from the North Slope of Alaska (NSA), as summarized in Appendix A. Fountain and Ohtake (1985) found mean INP abundance of $\approx 0.2\ L^{-1}$ at $-20\,°C$ at the surface there from August 1978 to April 1979. Prenni et al. (2007) measured similar $n_{INP}$ from aircraft, with a mean of $\approx 0.2\ L^{-1}$ in deposition and

condensation freezing modes over $\approx -8\,°C$ to $-28\,°C$. Elevated $n_{INP}$ (up to $\approx 40\ L^{-1}$) were measured in the temperature range between $\approx -14\,°C$ and $-30\,°C$ during the aircraft measurements along the NSA coast by Sanchez-Marroquin et al. (2023). While the authors found the INP source identification challenging (i.e., terrestrial, permafrost, maritime, biogenic, and/or a combination of any), their complementary aerosol particle composition and back trajectory results implied that

local and remote emissions and sinks of INP played an important role in the $n_{INP}$ variability.



Several aircraft-based studies documented that greater $n_{\mathrm{INP}}$ leads to more ice in arctic clouds. For example, Rogers et al. (2001) reported mean $n_{\mathrm{INP}}$ of up to 57 L$^{-1}$ in the examined temperature range between $-10\,°C$ and $-30\,°C$ during May 1998. High INPs in the NSA region were reported in a more recent research vessel study in the Chukchi Sea (Inoue et al., 2021). Based on offline freezing assay analysis, the authors measured up to $\approx 100$ L$^{-1}$ in the temperature range between $-7.5\,°C$ and $-29.5\,°C$. The observed high INP abundance during cold-air outbreak events was attributed to ocean mixing and associated sea spray emission of ice nucleation active organic substances. Over land, INP studies report ambient mineral dust to be a significant source of arctic INPs. For instance, high INP episodes were also seen in an Iceland study ($> 100$ L$^{-1}$ at $-26\,°C$; Sanchez-Marroquin et al. 2020) and from southern Alaska ($\approx 6$ L$^{-1}$ at $-26\,°C$; Barr et al. 2023), suggesting the importance of high latitude dust and other local terrestrial INP sources.

In contrast, Creamean et al. (2018a) reported lower $n_{\mathrm{INP}}$ at Oliktok Point, Alaska, about 250 km to the ESE of Utqiaġvik (formerly known as Barrow), Alaska. During March-May, 2017, they measured $n_{\mathrm{INP}}$ up to $\approx 4.4 \times 10^{-2}$ L$^{-1}$ for aerosol particles in the diameter range between 0.15 to 12 μm over the examined freezing temperatures. Creamean et al. (2018a,b) also found that the composition of aerosols from their study region varied, but it typically included terrestrial and/or maritime materials. Their source analysis postulates that bubble bursting and bacteria or fragments of marine organisms can act as the INP source from ice-free open water. Similarly, a ship-based study examining sea spray aerosol as the INP source over the central Bering Sea in summer 2012 found low abundance, up to $\approx 2.0 \times 10^{-2}$ L$^{-1}$ from $-12\,°C$ to $-20\,°C$ (DeMott et al., 2016). Low ambient $n_{\mathrm{INP}}$ has been found in arctic regions farther from Alaska, too. Creamean et al. (2022) reported $< 0.1$ L$^{-1}$ at $-25\,°C$ during the Multidisciplinary drifting Observatory for the Study of Arctic Climate (MOSAiC) expedition in the Central Arctic (September 2019 − October 2020). Similar to the MOSAiC finding, the offline freezing assay performed by Welti et al. (2020; W20 hereafter) showed $n_{\mathrm{INP}}(-28°C)$ of $\lesssim 0.2$ L$^{-1}$ from the PS 106 arctic expedition in the vicinity of Svalbard, Norway (May − July 2017). Continental dust during winter and marine biota from ice-free open water in summer were identified as the potential INP sources (Creamean et al., 2022; Irish et al., 2019a,b; Creamean et al., 2019; C19 hereafter).

Compiling eight previous INP studies from Alaskan, Canadian, and European arctic regions covering a wide range of freezing temperatures, Wilbourn et al. (2023) summarize abundance as spanning seven orders of magnitude ($\approx 10^{-5}$ to 70 L$^{-1}$). Because the INP abundance



is so variable, and most data thus far have been limited to brief campaigns, it is important to develop and analyze INPs statistically based on continuous, long-term, and finely-resolved measurements (Murray et al., 2021). This study represents one of the first efforts to elucidate

seasonality in the abundance of immersion mode active INPs using a single instrument, a Portable Ice Nucleation Experiment (PINE) chamber version 03 (PINE-03 hereafter). The PINE-03 was installed on the NSA near Utqiaġvik for multi-seasonal INP monitoring. In addition to a statistical analysis of $n_{INP}$, we combine the measurements with observatory data there to construct a parameterization for immersion freezing efficiencies of natural aerosols (i.e., $n_{INP}$ scaled to aerosol

abundance).

## 2. DATA & METHODS

### A. STUDY SITE AND PERIOD

Observations were made at the National Oceanic and Atmospheric Administration's (NOAA's) Barrow Atmospheric Baseline Observatory (71.32° N, 156.61° W, "BRW" hereafter), ~ 6 km

northeast of the town of Utqiaġvik. Our observing period began October 2021 and continued until May 2024 as the field component of Examining INP at NSA (ExINP-NSA), covering nearly 32 months. Here, we utilize data acquired from mid-October 2021 through December 2023.

Although the measurements at BRW are made over open tundra, there are large lagoons and numerous lakes in the vicinity, and the Arctic Ocean is less than 3 km to the north and east.

Because of its proximity to these bodies of water and the prevailing easterlies from the Beaufort Sea, BRW is perhaps best characterized as having an arctic maritime climate modulated by nearby sea ice conditions, but is also influenced by episodic atmospheric advection from the North Pacific (e.g., Cox et al., 2012, 2017). The BRW observatory was chosen for ExINP-NSA in order to collocate with NOAA's atmospheric baseline measurements, which include aerosol optical,

microphysical and chemical properties, and meteorology. To complement the current BRW capabilities, we experimentally characterized the INP abundance in association with the physicochemical properties of ambient aerosols. The findings are described throughout Sect. 3. The BRW site is equipped with well-characterized laminar flow stack inlets, and the air intake is about 40 feet (~ 12 m) above the ground level (AGL, Andrews et al., 2019). Moreover, at the

beginning of the field campaign, we conducted a complementary characterization of aerosol transmission efficiency through the inlet, and the result is reported in the Supplementary



Information (SI) Sect. S1. No corrections for particle losses or sampling conditions are applied to any aerosol data used in this report (see SI Sect. S1).

### B. INP CONCENTRATION MEASUREMENT

The PINE-03 system measures ambient $n_{INP}$ *in situ* using a simulated adiabatic expansion cooling method (Möhler et al., 2021). This system is a commercialized product, resulting in consistent operation amongst studies (Möhler et al., 2021; Knopf et al., 2021; Lacher et al., 2024; Wilbourn et al., 2024) compared to traditional INP monitoring devices that are typically custom-built by individual scientists. Besides relatively high measurement time resolution ($\lesssim$ 12 min), the

advantages of PINE-03 include (1) no substantial artifacts (e.g., no ice off of the vessel wall); (2) remote operation capability with minimum in-person maintenance or supervision requirements; and (3) fast turnover time to scan freezing temperatures in a wide range (Wilbourn et al., 2024).

The PINE-03 run is automated and continuous, reporting values approximately every 5 − 12 minutes. The system enables a simulation of atmospheric immersion freezing and deposition

ice nucleation depending on the vessel gas temperature and water saturation conditions, which can be controlled by the user via a digital interface. Previously, the ground-level immersion mode INP abundance was monitored for over 45 days by the same PINE-03 system during two field campaigns at U.S. Department of Energy (DoE) Atmospheric Radiation Measurement (ARM) program sites. These campaigns include Examining the Ice-Nucleating Particles from Southern

Great Plains (ExINP-SGP, Knopf et al., 2021) and Examining the Ice-Nucleating Particles from Eastern North Atlantic (ExINP-ENA, Wilbourn et al., 2024). Hence, our PINE-03 was tested in distinctly different environments (i.e., predominantly terrestrial and marine-influenced sites) to understand the properties of immersion-mode INPs with respect to the origin of air mass and ambient aerosol properties (i.e., number and surface area concentrations, as well as chemical

composition).

The PINE-03 system operates by cycles of "flush", "expansion", and "refill" modes. During the flush mode, ambient air is actively dried through a set of two Perma Pure dryers and is injected into the 10 L volume chamber with a flow rate of 2 LPM for 10 minutes. In the subsequent expansion mode, the sample gas temperature and pressure are reduced with $3\,\mathrm{L\,min^{-1}}$ of pump

flow rate to $800\,\mathrm{hPa}$ in the vessel to supersaturation with respect to both ice and water. This simulated adiabatic expansion typically lasts about one minute and triggers ice nucleation if INPs





are present in the sample. An optical particle counter (OPC; fidas-pine; Palas GmbH) deployed downstream of the chamber then detects particles exiting the chamber. Based on the optical size (typically > 10 μm in diameter), ice crystals can be separated from other particles (i.e., interstitial

aerosols and/or water droplets) and counted as immersion mode INPs. During the refill mode, filtered air is injected into the chamber for approximately a minute to precondition the vessel for the next run cycle.

To harmonize the datasets collected with different time intervals, the INP dataset was processed by averaging over 6 hours and synchronized to the same time scales following the

previous PINE-03 study led by Wilburn et al. (2024). In our typical chamber operation, the air gas set-point temperature is changed between −10 and −31 °C. The time resolution of such a temperature ramp was approximately 2 hours, and thereby the 6-hour time-averaged PINE-03 data represent $n_{INP}$ from three temperature ramps. Single PINE-03 'operation' typically lasts a day until the daily maintenance is performed. Therefore, a set of multiple temperature ramps was acquired

daily. The PINE-03 was cleaned daily by flushing filtered ambient dry air through the chamber until no particles were detected. We followed the other long-term chamber maintenance protocols as described in Wilbourn et al. (2024).

The highest freezing temperature for detecting INPs at NSA was −10.4 °C based on the original data acquisition time resolution. The PINE-03 system has a temperature uncertainty of ±

1.5 °C. A detection limit of PINE-03 is 0.2 L$^{-1}$ for individual expansion, which corresponds to a single INP detection per air volume assessed in a single expansion (≈ 3.4 L on average), and 0.02 L$^{-1}$ on a 6-hour time average basis, allowing summed air volume assessment specific to the ExINP-NSA condition. With this detection limit, a temperature-dependent Poisson error analysis was carried out in the field, which verified the statistical validity of the PINE-03 data below −16 °C

(see SI Sect. S2). Nonetheless, due to this upper temperature limit, we note that observed INPs do not necessarily represent INPs in near-surface clouds. Further details of the working principle of the PINE-03 system, as well as its calibration protocol and data, can be found in Möhler et al. (2021) and Wilbourn et al. (2024).

PINE-03 data flagging screens for operational issues. The most common problems include

an OPC malfunction or LabView data acquisition console disconnection. During ExINP-NSA, we rarely observed such issues (41 out of 1506 operations, 2.7%), and PINE-03 ran reliably with



scheduled maintenance periods. Operational flagging was assessed every cycle during measurements.

## C. AEROSOL DATA

### 1. AEROSOL NUMBER CONCENTRATION

Aerosol number concentrations ($n_{aer}$) were measured at BRW with a condensation particle counter (CPC; model 3010, TSI Inc.). The $n_{aer}$ was used to indicate the total aerosol particle abundance over the study period and to compute the INP-activated fraction ($IAF = n_{INP}(T)/n_{aer}$). In addition, another CPC (model 3772, TSI Inc.) was operated at the adjacent DoE-ARM site as part of the

NSA Aerosol Observing System. Both the 3010 and 3772 CPC have a 10 nm minimum cut size. Over our ≈ 2-year study period, similar $n_{aer}$ was measured by BRW-CPC (median of 156.3 cm$^{-3}$) and ARM-CPC (179.0 cm$^{-3}$) for non-screened datasets. Although BRW-CPC reads slightly lower than ARM-CPC based on 6-hour time-averaged medians, the Pearson correlation coefficient, $r$, between two CPC datasets is high ($r \approx 0.9$). All $n_{aer}$ presented here are from the BRW-CPC. To

make all the data from instruments that have different sampling times comparable, all online datasets discussed in this study were averaged over 6-hour periods.

### 2. SURFACE AREA CONCENTRATION

We estimate the aerosol surface area concentration ($S_{aer}$) at volume standard temperature and pressure (273.15 K and 1013.25 hPa) using NOAA's aerosol scattering coefficients measured by

an integrating nephelometer (Model 3563, TSI Inc.). Aerosol scattering coefficients from the nephelometer are reported in units of inverse megameters (Mm$^{-1}$). The application of the nephelometer data to calculate the aerosol surface areas has been demonstrated in prior studies in marine conditions (DeMott et al., 2016; Wilbourn et al., 2024). Aerosol scattering coefficients at three wavelengths (450, 550, and 700 nm) were continuously measured by the nephelometer,

which was operated under low humidity conditions (< 40 % relative humidity). $S_{aer}$ values are computed by scaling aerosol scattering coefficients at 450 nm ($b_{sp}^{450}$) by a factor of 4 and normalizing the scaled number to Q using the following equation (Moore et al., 2022):

$$S_{aer} = 4\frac{b_{sp}^{450}}{Q}, \qquad [1]$$

where Q is an effective aerosol scattering efficiency. The monthly averaged coarse mode (i.e.,

PM$_{10}$ − PM$_1$) Q value of 2.37 (± 0.04 standard deviation) derived during clean marine conditions





at El Arenosillo, Spain, is considered a representative Q and used in this study. More details are discussed in SI Sect. S3. We use $S_{aer}$ to assess particle surface area and to compute ice nucleation active surface site density, $n_s(T) = n_{INP}(T)/S_{aer}$.

### 3. BLACK CARBON MASS CONCENTRATION

Black carbon mass concentration ($m_{BC}$) was estimated for the $PM_{10}$ size range based on the Continuous Light Absorption Photometer (CLAP, Ogren et al., 2017). The CLAP is a filter-based instrument that uses Beer's law to relate the change in optical transmission through a filter caused by particle deposition to the light absorption coefficient of deposited particles. Aerosol absorption coefficients from the CLAP are also reported in units of $Mm^{-1}$. Measured mass absorption cross-

section values for freshly generated black carbon fall within a relatively narrow range of $7.5 \pm 1.2\ m^2\ g^{-1}$ at 550 nm (Bond et al., 2013). This assumption of uniform aerosol composition may introduce uncertainties in information derived from CLAP data, which represents a limitation of this study, as few natural aerosol populations have uniform composition. Here, $m_{BC}$ ($ng\ m^{-3}$) was estimated by dividing the absorption at 528 nm by the estimated mass absorbing cross-section

of $7.5\ m^2\ g^{-1}$ (Zheng et al., 2018; Bond et al., 2013).

### 4. PARTICLE SULPHATE AND NITRATE MASS CONCENTRATION

Ambient mass concentrations of major arctic haze tracers, such as non-sea salt (nss) $SO_4^=$ and aerosol $NO_3^-$, were measured using filter samples of atmospheric aerosols collected at BRW for subsequent ion analysis (Quinn et al., 2002; 2000; 1998). We calculated nss particle sulfate mass

concentration by relating total $SO_4^=$ ion mass concentration to the mass concentration of reference species, such as sodium, in seawater ([nss $SO_4^=$] = [$SO_4^=$] − (0.252 x [Na]) (Keene et al., 1986). While both submicron and supermicron particle data are available, supermicron data availability had very limited temporal resolution (minimum 7 days, maximum 28 days). Therefore, we used the submicron dataset to represent arctic haze tracers. It is worth noting that submicron $SO_4^=$ and

$NO_3^-$ were the predominant contributors to total mass of the submicron ions (a factor of 4 − 5 more compared to supermicron mass where sea salt is the dominant species) for periods when both datasets were available. We also note that sampling resolution of this offline ion analysis data is much longer than 6 hours (minimum 24 h, maximum 96 h), and the sampling interval varied with season. Therefore, the 6-hour time averaging protocol was not applied for this offline data, and we

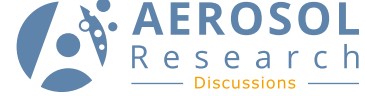

report the ion concentration data in its native time resolution. The filter measurements only sample
when air is coming from the clean air sector (see next section).

## 5. AEROSOL DATA FLAGGING

NOAA's aerosol data protocol flags data as contaminated when the measured wind direction (WD)
is aligned with the town of Utqiaġvik (i.e., 130° < WD < 360°). Thus, aerosol data from the wind

direction of Utqiagvik are automatically flagged. The clean air sector at BRW is to the east (0° <
WD < 130°). The full flagging method is described in Sheridan et al. (2016). Briefly, in addition
to the wind direction criterion, CPC spikes, notable contaminations identified by instrument
mentors, and abnormally low wind speed time periods are integrated in the flagging algorithm.
This method is consistent across the NOAA observatories and varies only by clean air sector

definition. NOAA provides flagging information for the aerosol data for every minute. NOAA's
aerosol data flagging was synchronized to the PINE-03 data acquisition interval.

At BRW, easterly winds and emissions from the Prudhoe Bay oil field can impact
measurements (Kolesar et al., 2017; Creamean et al., 2018a). However, the oil field is located ≈
300 km east of Utqiaġvik. Because we cannot easily segregate Prudhoe Bay emissions from other

local emissions, data coinciding with easterly winds are not flagged in this study. Although $m_{BC}$
could be used as a proxy of potential oil field emission only when the wind was from the clean
sector (Sect. 3B and Fig. 3), it could also be due to recirculation of air masses containing emissions
from the nearby community of Utqiaġvik.

### D. METEOROLOGICAL & AIR MASS DATA

Local meteorology, including wind speed, wind direction, temperature, and relative humidity,
were from BRW (see Data Availability). We used temperature data at 10 m AGL, which is nearest
to the stack inlet height. To compare with INP data, which are collected at different timescales,
meteorological datasets were also averaged over 6-hour time periods. Visibility and time-averaged
cumulative precipitation observations are not made at BRW, but are reported at the Wiley Post-

Will Rogers Memorial Airport (ICAO: PABR) (71.285° N, 156.769° W) located ~ 7 km southwest
of our field site.



# 3. RESULTS

### A. METEOROLOGY

Figure 1 displays meteorological data collected at BRW and their seasonality from October 19th, 2021, to December 31st, 2023. The overall median temperature during our study period was −7.7 °C. There is a pronounced

seasonality in temperature with a summer maximum of 19.7 °C and winter minimum of −37.2 °C. The median relative humidity (w.r.t. water) measured at BRW was 84.2 %. Particularly in winter, the air is typically near-

saturation or supersaturation w.r.t. ice.

   The seasonal average of measured visibility during our study ranged from 6.9 to 8.4 km without any distinct seasonal patterns. Both the lowest and highest average visibilities

were measured in winter (low in 2022 and high in 2023). It is noteworthy that the 6-hour

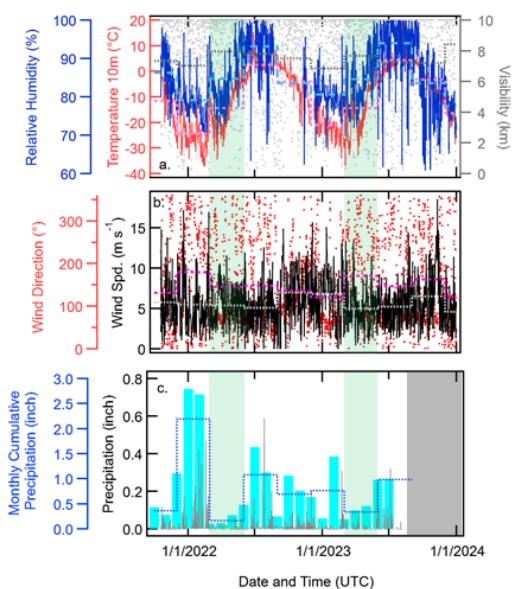

Figure 1. The time series of the 6-hour average temperature, relative humidity, and visibility (a), as well as wind properties (b). Panel (c) displays the 6-hour average precipitation and monthly cumulative precipitation amounts. Dashed lines in each panel are mean seasonal values of individual measurements and the green shaded area represents spring. Precipitation data from mid-August 2023 was not available, which is indicated by the grey shaded area. The relative humidity data from late August to early December 2022 is also missing.

average visibility fluctuated throughout 2021 − 2023. The observed visibilities are seasonally consistent but occasionally variable, implying a strong influence of localized events, such as blowing dust, blowing snow, haze, fog, and sea spray (e.g., Chen et al., 2022; Quinn et al., 2007;

DeMott et al., 2016).

   Seasonal wind roses are plotted in Fig. 2. Median annual wind speed (± standard error) at BRW was $5.2 \pm 1.6$ m s$^{-1}$. During fall − winter, the seasonal average wind speed ranged from 6.4 m s$^{-1}$ (SON) to 5.6 m s$^{-1}$ (DJF). During spring − summer, the seasonal average wind speed was similar (5.2 m s$^{-1}$). The maximum wind speed of 18.6 m s$^{-1}$ was measured in November 2023.

Although there was variability in wind direction measured at BRW, northeasterly winds prevailed as expected throughout the study period, which is predominantly from the clean air sector upwind of nearby settlements.



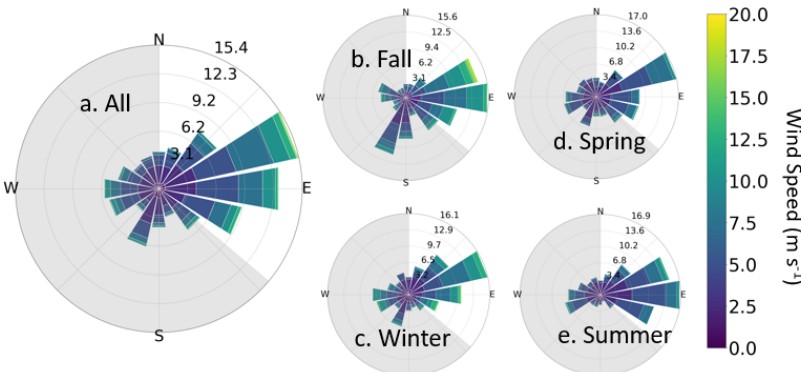

Figure 2. The wind speed and direction distributions during the ExINP-NSA (October 2021 − December 2023) are
shown in the wind rose plot (a). The color scale of wind roses represents the wind speed observed at ground level (11
meters above sea level). Panels b − e show the wind roses from different seasons; fall (b), winter (c), spring (d), and
summer (e). The grey shaded area represents the flagged wind direction (130° < WD < 360°) indicating potential
contamination from the nearby community of Utqiaġvik.

The median value of monthly cumulative precipitation (± standard error) measured at the
BRW site was $14.2 \pm 4.1$ mm. As seen in Fig. 1c, biannual maxima of measured precipitation in
winter (37.6 mm) and summer (26.4 mm) were found in 2021 – 2023; the lowest amount of
precipitation occurred in spring (mean 6.6 mm).

**B. Aerosol Abundance**

Figure 3 shows time series plots of $n_{aer}$ and $S_{aer}$, black carbon mass, and submicron ion mass
concentrations of arctic haze tracers. The total $n_{aer}$ ($cm^{-3}$, shown with black dots) is plotted at 6 h
averaged intervals. The overall median $n_{aer}$ (± standard error) during October 2021 − December
2023 was $156.3 \pm 8.1$ $cm^{-3}$, while the seasonal average $n_{aer}$ was highest in summer
($589.1 \pm 54.2$ $cm^{-3}$) and lowest in winter ($227.2 \pm 18.9$ $cm^{-3}$). On average, spring also exhibited a
relatively high $n_{aer}$ of $431.4 \pm 44.0$ $cm^{-3}$, implying an influence of arctic haze (Quinn et al., 2007).
Seasonal averages from this study are consistent with a long-term trend of monthly geometric
means of condensation nuclei measured at BRW from 1977 to 1994 with an annual cycle of typical
summer maxima and winter minima (Polissar et al., 1999).





Estimated $m_{BC}$ values are also shown in
Fig. 3a. With a median ± standard error value of
13.2 ± 3.4 ng m$^{-3}$, a strong winter maximum is
apparent (up to 92.2 ± 3.9 ng m$^{-3}$) consistent with
previous reports of seasonality of absorbing
aerosol at BRW (e.g., Polissar et al., 1999;
Delene and Ogren, 2002; Schmeisser et al.,
2018). The highest $m_{BC}$ was also observed during
spring with ~ 40 ng m$^{-3}$ on average. Previously,
Barrett and Sheesley (2017) reported a peak
elemental carbon (EC) mass concentration ($m_{EC}$)
of ~ 100 ng m$^{-3}$ measured at the ARM-NSA
facility in February 2013. The authors identified
fossil fuel combustion via transport as a
significant source of ambient organic carbons,
accounting for > 60% of mass, during their year-
round study period from the summer of 2012.
Moffett et al (2022) measured low $m_{EC}$ near
Utqiaġvik during summer, suggesting biomass
burning and wildfire contribution as a minor
source of EC.

Figure 3. The 6-hour average total particle concentration ($n_{aer}$, cm$^{-3}$, shown with black dots) and the mass concentration of black carbon ($m_{BC}$, ng m$^{-3}$, red crosses) (a). The time series of the 6-hour average total surface area concentration ($S_{aer}$, m$^2$ L$^{-1}$, shown with black crosses) (b). Submicron NO$_3^-$ and nss SO$_4^=$ ion mass concentrations (c). The error bars in (a) and (b) represent standard errors of each measurement. The uncertainties in (c) are reported in the chemical data. A dashed horizontal line in each panel represents the seasonal mean of individual measurements, and the green shaded area represent the Arctic spring during our study periods.

The median $S_{aer}$ (± standard error) at
BRW was $1.2 \times 10^{-9} \pm 8.4 \times 10^{-11}$ m$^2$ L$^{-1}$. Seasonal variability in $S_{aer}$ is shown in Fig. 3b, with seasonal average maxima and minima found in winter 2021 (DJF; $3.2 \times 10^{-9} \pm 1.8 \times 10^{-10}$ m$^2$ L$^{-1}$) and summer 2022 (JJA; $1.3 \times 10^{-9} \pm 1.3 \times 10^{-10}$ m$^2$ L$^{-1}$), respectively. The estimated median single particle surface area (i.e., $S_{aer}/n_{aer}$) from BRW (< 0.02 μm$^2$) is substantially smaller than at ARM-SGP (1.4 μm$^2$) and ARM-ENA (0.05 μm$^2$) derived from Wilbourn et al. (2024), suggesting a predominance of small particles at BRW. We note that $S_{aer}$ is derived by means of nephelometer measurements at both BRW and ARM-ENA. *In-situ* coarse aerosol size distribution measuring instruments, such as an optical particle counter and an aerosol particle sizer, were not operational during any of our campaigns.

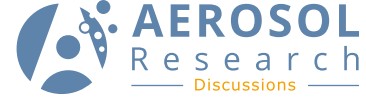

The min-max ranges of nss $SO_4^=$ and $NO_3^-$ at BRW are 0.003 – 2.2 and 0.005 – 1.2 µg m$^{-3}$, respectively, during our field study. Clear seasonal cycles were found for arctic haze tracers, including nss $SO_4^=$ and $NO_3^-$ (Fig. 3c). With a median mass concentration of 0.2 ± 0.02 µg m$^{-3}$, the maximum mass concentration of nss $SO_4^=$ was found in spring on average (0.4 ± 0.03 µg m$^{-3}$). Likewise, $NO_3^-$ also had the highest seasonal average of 0.1 ± 0.01 µg m$^{-3}$ in spring. The observed

spring maxima and seasonality in particulate nss sulfate and nitrate mass concentrations can primarily be attributed to the long-range transport of arctic haze (Quinn et al., 2007). We also note that, because these aerosol composition values are for submicron soluble aerosol, these chemistry measurements may not directly relate to INPs, as INPs preferentially involve insoluble supermicron particles (e.g., Mason et al., 2016).


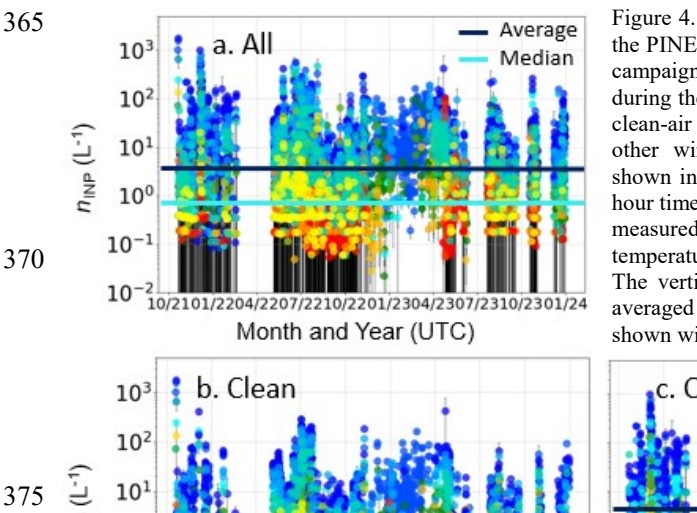

Figure 4. INP concentrations ($n_{INP}(T)$) measured at BRW with the PINE-03 system. The 'all' dataset collected throughout the campaign is shown in (a). The segregated datasets collected during the 'clean' periods when wind directions were from the clean-air sector (clean data) and 'contaminated' periods for other wind directions (presumably contaminated data) are shown in (b) and (c), respectively. Each point represents a 6-hour time-averaged concentration. The color scale indicates the measured freezing temperature. Individual data points are temperature binned for 1 °C and rounded to the closest integer. The vertical error bars represent the standard error of time-averaged data. The campaign mean and median $n_{INP}(-25°C)$ are shown with dark blue and cyan lines, respectively.


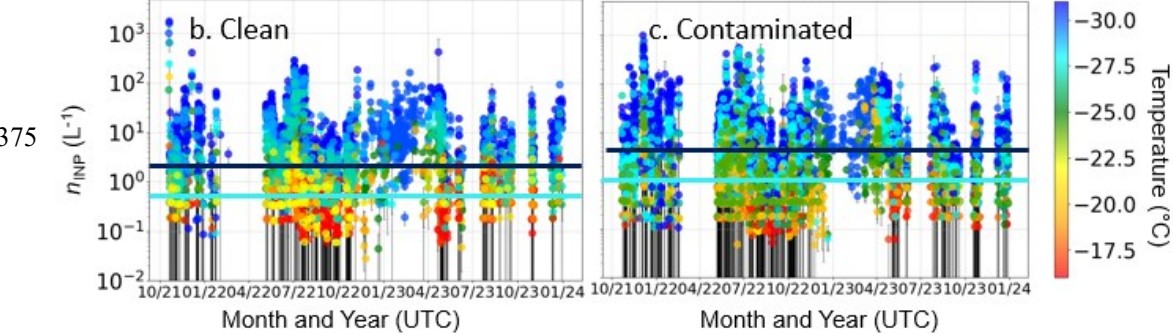


## C. ICE-NUCLEATING PARTICLE ABUNDANCE

Shown in Fig. 4 is the comparison of online $n_{INP}(T)$ based on (a) the 'all' dataset (i.e., all valid measurements retained); (b) 'clean' data subset as determined following the standard BRW wind protocols and removing flagged PINE-03 data for operational issues; and (c) 'contaminated' subset following the wind and PINE-03 data screening protocols. The time series of 6 h averaged $n_{INP}(T)$ from BRW with a temperature resolution of 1 °C is shown in each panel, with different colors



scaling to the freezing temperature between −16 °C (red) and −31 °C (blue). For the 'all' dataset, the $n_{INP}(T)$ data are displayed with a total of 14,318 data points of 6 h averaged $n_{INP}(T)$ collected during our study period. The data gaps in spring 2022, summer 2023, and fall 2023 seen in Fig. 4a are due to maintenance, as required every 3 − 4 months (see Wilbourn et al., 2024; SI Sect. S5).

       For freezing temperatures from −16 to −31 °C, clean $n_{INP}(T)$ data show the lowest average

(± standard error). As shown in Fig. 4, $n_{INP}(-25°C)$ values ± standard errors are $3.6 \pm 1.2 \, L^{-1}$, $2.1 \pm 0.6 \, L^{-1}$, and $4.6 \pm 1.5 \, L^{-1}$ for all, clean, and contaminated datasets, respectively. Likewise, the medians of $n_{INP}(-25°C)$ are similarly sorted with $0.8 \pm 0.4 \, L^{-1}$, $0.6 \pm 0.2 \, L^{-1}$, and $1.1 \pm 0.5 \, L^{-1}$ for all, clean, and contaminated datasets. As anticipated, the contaminated dataset exhibited a higher mean than the others, likely reflecting the influence of emissions from Utqiaġvik. The distribution

of $n_{INP}(T)$ is skewed due to the occurrence of positive extremes. Thus, we report the median in addition to the mode.

### D. FREEZING EFFICIENCIES

Figure 5 shows the 6-hour average $n_{INP}(T)$, *IAF* (i.e., $n_{INP}(T)/n_{aer}$), and $n_s(T)$ (i.e., $n_{INP}(T)/S_{aer}$) at selected temperatures (−20, −25, and −30 °C). A noticeable difference between $n_{INP,all}$ and $n_{INP,clean}$

is seen in Fig. 5. Typically, we observe that $n_{INP,all}$ exceeds $n_{INP,clean}$ as the all/clean ratio is typically > 1 (Fig. 5g − i). In winter, the ratio is especially high. During this time, southwesterlies, presumably contaminated by recirculated emissions from the town, contain abundant INPs. In winter 2021, the seasonal $m_{BC}$ of 92.2 ng m$^{-3}$ is higher than the overall average $m_{BC}$, which indicates the impact of Utqiaġvik emissions (e.g., fuel burning). We note that a seasonal average

$m_{BC}$ of 21.5 ng m$^{-3}$ in winter 2022 is lower than the overall average $m_{BC}$, suggesting that local emissions may not have made a prominent contribution to $m_{BC}$ observed at BRW in winter 2022 and that BC is in part from long-range transport as suggested by previous studies (Barrett and Sheesley, 2017; Moffett et al., 2022). High INP abundance and freezing efficiencies not associated with local emissions were observed in spring 2023. This coincided with a large temporal change

in ambient temperature and minimal seasonal precipitation (Fig. 1), as well as observed high concentrations of arctic haze tracers (Fig. 3). Hence, this high INP episode may have been triggered by a combination of factors. Average *IAF*s at −20, −25, and −30 °C are similar between the all and clean datasets ($1.7 \times 10^{-6}$ − $1.1 \times 10^{-4}$). Conversely, $n_s(T)$ exhibits a slight deviation between the two datasets with 'clean' having a lower average $n_s(T)$ of $\approx 3.2 \times 10^8 \, m^{-2}$ to $\approx 1.1 \times 10^{10} \, m^{-2}$ than



the 'all' dataset ($\approx 9.4 \times 10^8 \, \text{m}^{-2}$ to $\approx 1.6 \times 10^{10} \, \text{m}^{-2}$). In order to relate our results to BRW baseline

aerosol measurements and previous literature, 'clean' sector data are used for further analysis in

this study.

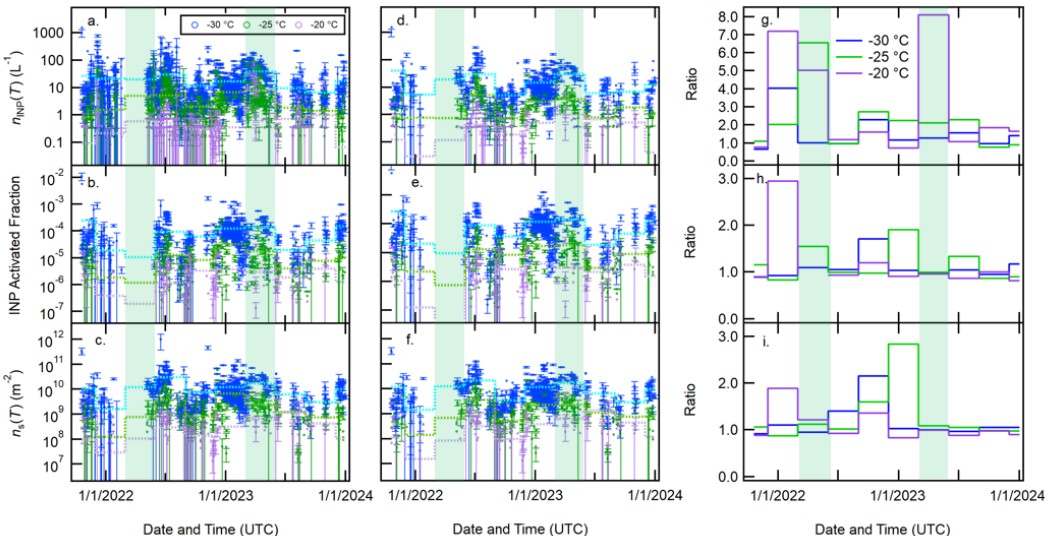

Figure 5. The 6-hour time-averaged $n_{\text{INP}}(T)$, $IAF$, and $n_{\text{s}}(T)$ at selected temperatures for the 'all' dataset (a − c) and screened 'clean' data subset (d − f) at BRW. Panels (g − i) show the ratio of all/clean data seasonally. Dashed lines represent seasonal average values for the measured periods. The $n_{\text{INP}}$ error bars represent standard errors for individual 6-hour averaged data points. The standard errors for IAF and $n_{\text{s}}$ were computed by taking the square-root of the total relative standard errors for individual 6-hour averaged data points. Green shaded area represents the arctic spring period.

A series of histograms displaying probability densities and relative frequency of 6-hour

averaged $n_{\text{INP}}(T)$ and $n_{\text{s}}(T)$ data from PINE-03 are shown in Fig. 6 with a temperature resolution

of 1 °C for BRW. As seen, the mode $n_{\text{INP}}(T)$ and $n_{\text{s}}(T)$ are reasonably comparable to our average

$n_{\text{s}}(T)$ for data with the given bin-resolved data density ($n > 224$) despite some inclusion of outliers

at low $n_{\text{INP}}(T)$ and $n_{\text{s}}(T)$. For the $n_{\text{s}}(T)$ distributions, fitted $n_{\text{s}}(T)$ values from this study are also

superimposed on each histogram to show reasonable agreement with the average values of the log-

normal $n_{\text{s}}(T)$ distribution. Seasonal breakdowns of the $n_{\text{INP}}(T)$ and $n_{\text{s}}(T)$ histograms are shown in

SI Figs. S2 and S3 (SI. Sect. 4).






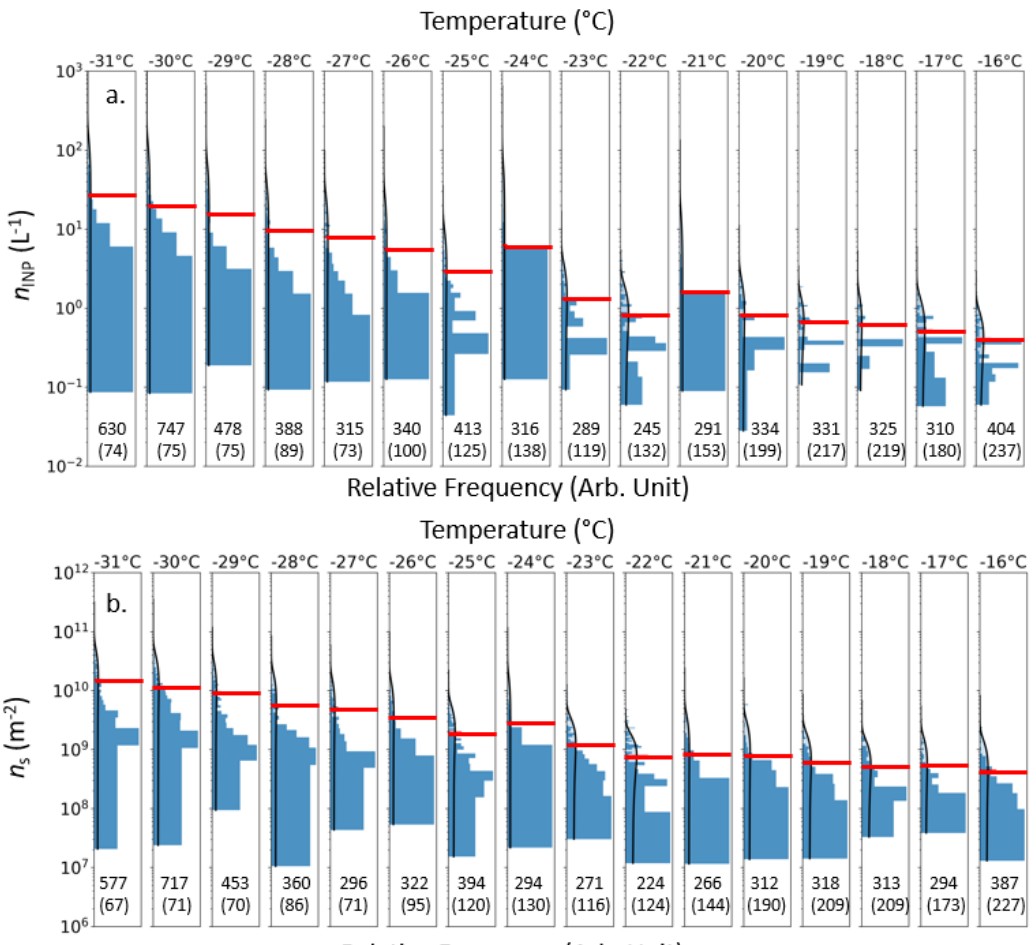

Figure 6. Histograms of the PINE-03-based $n_{INP}(T)$ and $n_s(T)$ Gaussian distribution with one degree temperature binning are shown in (a) and (b). The 'clean' data were used to generate this figure. The data covers a statistically validated freezing temperature range (−16 to −31 °C) for October 2021 − December 2023. Individual data densities are shown at the bottom of each column, and zero INP number counts, included in time-averaged $n_{INP}$ calculation, are shown in parentheses. Relative frequencies (Arbitrary Unit) for each degree are shown at the bottom of each sub-panel. Red horizontal lines in each relative frequency distribution sub-panel represent the average. The Gaussian log-normal fit is shown for each degree of binned data (black lines).

Figure 7 shows 6-hour averaged PINE-03-measured $n_{INP}$ and $n_s$ data from BRW as a function of freezing temperatures (1 °C resolution) as box plots (a − b). Clean data were used to generate Fig. 7 while Fig. S4 is based on 'all' data for comparison (SI Sect. S5). Also shown in Fig. 7a are previously reported $n_{INP}(T)$ data collected from or near the North Slope of Alaska (see Sect. 1D and references therein). The data collected in this study are generally comparable to data

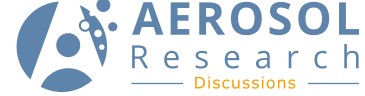

presented in Barr et al. (2023; B23), Inoue et al. (2021; I21), Sanchez-Marroquin et al. (2023; S-M23), and Prenni et al. (2007; P07) as their data overlap with our $25^{\text{th}} - 75^{\text{th}}$ percentile $n_{\text{INP}}(T)$ data in one temperature bin at least. On the other hand, the $n_{\text{INP}}(T)$ range for some studies is much lower than the $n_{\text{INP}}(T)$ range of ExINP-NSA, potentially due to differences in INP sources that those studies investigated (e.g., sea spray aerosols without sea ice coverage). Figure 7b shows the $n_{\text{s}}(T)$ data, as well as associated exponential fits. Following Li et al. (2022) and Wilbourn et al. (2024), we computed $n_{\text{s}}(T)$ parameterizations that fit the average values of the log-normal $n_{\text{s}}(T)$ distribution as a function of freezing temperatures as follows:

$$n_s^{avg}(T) = exp\left(24.250 \times exp\left(-exp\left(0.060 \times (T + 9.700)\right)\right) + 4.995\right) \qquad r = 0.99$$

$$-31\,°C \leq T \leq -21\,°C. \qquad\qquad [2]$$

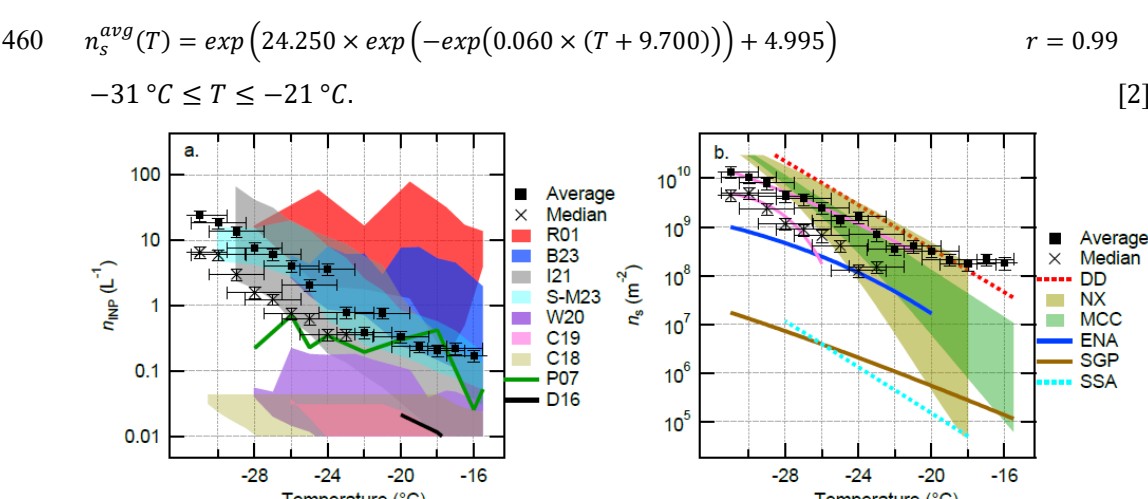

Figure 7. Box plots of the PINE-03 based $n_{\text{INP}}(T)$ (a) and $n_{\text{s}}(T)$ (b) data from BRW in 1 degree temperature bins for a statistically validated freezing temperature range (−16 to −31 °C). The 'clean' data were used to generate this figure. Boxes represent average (black solid symbol) and median (black open symbol) statistics. The color-shaded area in (a) shows the maximum and minimum $n_{\text{INP}}(T)$ measured by previous INP studies at or in the proximity of BRW (see Table A1 and Sect. 1 for references). The reference $n_{\text{s}}(T)$ data in (b) are adopted from W24 (Wilbourn et al., 2024 and references therein) for SGP, ENA, desert dust, sea spray aerosol, illite NX, and microcrystalline cellulose. Pink lines are fits to BRW data from this study. The uncertainties in $n_{\text{INP}}(T)$ and $n_{\text{s}}(T)$ are also adopted from W24.

The parameterization offered in this study is limited to $\leq -21$ °C. Below −21 °C, a constant increase in $n_{\text{s}}(T)$ towards low freezing temperature is seen, whereas a plateau of high $n_{\text{s}}(T)$ is found between −21 °C and −16 °C, at which our INP data are validated within errors discussed in SI Sect. S2. However, we cautiously note that the flattening of the concentrations warmer than −21 °C is a spurious result mainly due to the instrument resolution. The total effective sampling volume is a combination of the chamber size, the number of sampling cycles that are averaged, and the pressure to which the chamber is filled. The minimum resolvable INP value is 0.02 L$^{-1}$ on a 6-



hour time average, but non-time-averaged minimum $n_{INP}$ detection limit is in fact ~ 0.3 $L^{-1}$. This floor approximately intersects where the data remains remarkably steady across the whole

temperature range and a value where extrapolation of the functional relationship of concentration and temperature would suggest is crossed near −20 °C (i.e., close to the beginning of the flattening). We would expect a loss of sensitivity to result in an undercounting of values as mostly 0s are averaged into data. Nevertheless, it appears likely that the flattening is a consequence of the resolution floor of the system and its operational configuration at BRW.

The comparison between $n_s(T)$ data from this study and reference spectra shown in Fig. 7b reveals that immersion freezing efficiencies of aerosols collected at ground level at BRW are equivalent to, or higher than, desert dust studied in Ullrich et al. (2017) above −20 °C. This outcome was expected as the aerosol population at BRW is presumably not purely composed of desert dust. Indeed, many previous studies suggest the potential influence of highly active biogenic

INP in the region (Inoue et al., 2021; Creamean et al., 2022). While a partial overlap of our $n_s(T)$ with illite NX (mineral dust proxy) and microcrystalline cellulose (MCC; non-proteinaceous organic surrogate) spectra are seen in a few temperature bins in the middle range (i.e., −27 °C < $T$ < −19 °C), reference spectra of these compositions cannot solely explain the overall $n_s(T)$ trend from BRW. The sea spray aerosol (SSA) $n_s(T)$ parameterization spectrum from McCluskey et al.

(2018) shows a less active trend and is not comparable to the BRW data, implying aerosols collected at BRW are different from SSAs seen in McCluskey et al. (2018) and perhaps predominantly composed of nss and non-SSA compounds. The link between these chemical compounds to INP is not straightforward. Without detailed size-dependent composition and ice residual chemistry data, further discussion cannot be made in this study. It is also worth noting that

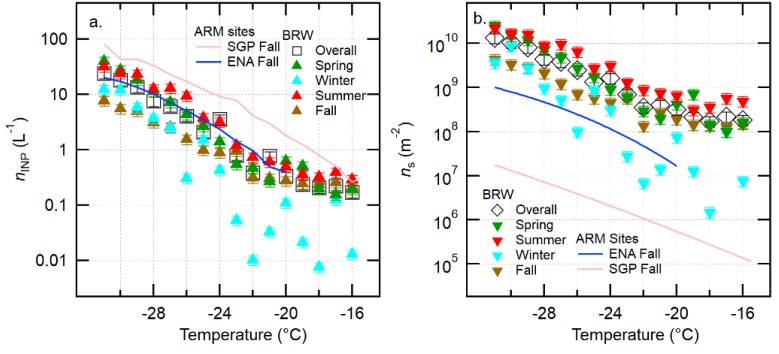

Figure 8. Seasonal breakdowns of the PINE-03 based $n_{INP}(T)$ and $n_s(T)$ data are shown in (a) and (b), respectively. The uncertainties in $n_{INP}(T)$ and $n_s(T)$ are also adopted from W24.



a substantial portion of the PINE measurement period was during winter when the adjacent ocean was capped by sea ice. Figure 8 below and Fig. S5 (SI Sect. 6) show seasonal $n_s$ parameterizations.

Shown in Fig. 8 is seasonality of 6-hour averaged PINE-03-measured $n_{INP}(T)$ and $n_s(T)$ data from BRW. When comparing seasonally-averaged $n_{INP}(T)$ values (Fig. 8a), it is notable that

$n_{INP}(T)$ in spring and summer at BRW is consistently higher than $n_{INP}(T)$ from other seasons. The observed difference in $n_{INP}(T)$ can be in part attributed to arctic haze episodes that occur during arctic spring (Rogers et al., 2001; Quinn et al., 2007) and local sediment exposure to air after springtime melt (Cox et al., 2019). Fall $n_{INP}(T)$ data from BRW in comparison to $n_{INP}(T)$ spectra from mid-latitude sites (i.e., SGP and ENA) in the same season suggest that INP abundance is

lowest in the Arctic (at least for fall). The maritime $n_{INP}(T)$ represented by the ENA measurements is consistently higher than the fall data from BRW and lies toward the upper bound of the overall BRW data. We note that relative abundance of aerosols at ENA is on average more than twice as high as observed at BRW for our study period. Continental INPs from SGP exceed BRW $n_{INP}(T)$ below −20 °C. It is worth noting that the high variability in the BRW winter data is partially due

to the high frequency of zero INP counts collected in this season ($\approx 66\%$) as compared to other seasons ($\approx 21 - 23\%$; Fig. 6). PINE-03 is designed to utilize ambient moisture to saturate the chamber during expansion cooling and for maintaining the chamber dew point temperature above freezing temperature. Dry winter conditions often lowered dew point and hindered INP measurements. Regardless, patterns in $n_{INP}(T)$ and $n_s(T)$ can still be compared as representative of

each season because our temperature-binned $n_{INP}(T)$ and $n_s(T)$ data offer at least 22 and 21 data points in each bin (see SI Figs. 2 and 3).

Seasonal variability in $n_s(T)$ is obvious in Fig. 8b. In general, two data subsets (i.e., higher $n_s(T)$ in spring and summer and lower $n_s(T)$ in fall and winter than the overall data) define the $n_s(T)$ characteristics from this study at low temperatures. Surprisingly, BRW $n_s(T)$ exceeds SGP and

ENA $n_s(T)$ values by at least one order of magnitude across the freezing temperatures analyzed in this study, suggesting there are unique INP properties in the region.

Correlations between detectable $n_{INP}$ at selected temperatures (i.e., −20, −25, and −30 °C) vs. measured variables averaged for 6 hours suggest the following: (1) $S_{aer}$ and $m_{BC}$ are well correlated ($r = 0.7$, $p < 0.05$), indicating some BC was externally mixed and available on aerosol

surfaces at BRW during the study period and, (2) at a freezing temperature of −25 °C, there is a positive correlation between $n_{INP}$ and precipitation amount ($r = 0.7$, $p < 0.05$; $N = 68$), which could



suggest a contribution of hydrometeors to $n_{INP}$, potentially derived locally in part by blowing snow (Chen et al., 2022). However, the correlation between precipitation amount and INP abundance for other temperatures is weak ($|r| \lesssim 0.2$, $p < 0.05$). Therefore, the direct relationship between INP and precipitation at BRW is not conclusive. It is noteworthy that our previous study with PINE-03 from a mid-latitude continental setting showed $n_{INP}$ values decreased immediately after precipitation events while IAF and $n_s$ remained consistent (Wilbourn et al., 2024).

While 6-hour time-averaged data is unavailable from the ion chromatography filter measurements (Sect. 2C.4), seasonal means of nss $SO_4^=$ correlate well with $NO_3^-$ ($r = 0.7$), wind direction ($r = 0.7$), $S_{aer}$ ($r = 0.7$), and $m_{BC}$ ($r = 0.9$). These correlations imply that arctic haze coincidentally delivers nss $SO_4^=$, BC, and $NO_3^-$ with large particle surface areas. On the other hand, nss $SO_4^=$ shows a reciprocal relation with temperature ($r = -0.7$), attributed to the winter − spring dominance of arctic haze. Furthermore, seasonal mean nss $SO_4^=$ weakly correlates with seasonal precipitation amount ($r = 0.5$). This implies that wet deposition during arctic haze may contribute to observed high nss $SO_4^=$ via evaporation and/or sublimation of the precipitation near the surface.

### E. AIR MASS TRAJECTORIES AND PARTICLE ABUNDANCE

The $n_{INP}$ observations are positively correlated with a regional climate index ($r \approx 0.4$ at −31 °C) that encodes the juxtaposition of the Aleutian Low and the Beaufort High (Cox et al., 2019). This indicates that higher INP concentrations tend to be associated with air advecting northward through the Bering Strait before dispersing eastward over the NSA during periods when the dominant easterlies of the Beaufort High are weak or reversed. In this section, we further analyze the source regions using back trajectories.

Table 2 lists periods of high- and low-INP episodes and associated $n_s(T)$ parameters found at BRW. To find these episodes, we first identified periods of high- and low-INP episodes and associated $n_s(T)$ parameters found at BRW during our study. Since $n_s(T)$ accounts for both INP and aggregate aerosol properties, we use it as a representative ice nucleation efficiency index to select high- or low-INP periods in this study. High INP episodes were identified by extracting periods when the 6-hour time-averaged $n_s$ values exceed their 75[th] percentile values using three reference temperatures, −20, −25, and −30 °C. In contrast, low INP episodes are represented by times when $n_s$ at the three temperature values was below the 25[th] percentile. We made two subsets of high- and low-INP episodes; one where all three temperatures had to exceed the percentile



thresholds ('all three $T$s') and another with the same thresholds but where the sample qualified if 'any' of the three examined temperatures met the threshold. For the former case, we identified 15

high INP episodes and 15 low INP episodes. For the latter case, we identified 291 data points as being in a high INP period and 364 as being in a low INP period (SI Table S3).

Table 2. List of high- and low-INP periods from BRW for subsets of 'all three $T$s' data. *Clean data (A fully extended table is available in supplemental materials)

| Data | ID | Date & Time (UTC) | $n_s$(m$^{-2}$) | | |
|---|---|---|---|---|---|
| | | | −30 °C | −25 °C | −20 °C |
| All Three $T$s | 1 | 5/9/2022 0:00 | 2.1E+10 | 2.2E+09 | 5.7E+08 |
| High | 2 | 5/31/2022 18:00 | 2.4E+10 | 2.8E+09 | 7.2E+08 |
| INP | 3 | 6/20/2022 6:00 | 1.6E+10 | 2.7E+09 | 3.3E+08 |
| n = 15 | 4 | 6/20/2022 0:00 | 4.0E+10 | 6.9E+09 | 3.3E+08 |
| | 5 | 6/16/2022 6:00 | 3.9E+10 | 7.6E+09 | 1.3E+09 |
| | 6 | 6/25/2022 0:00 | 5.9E+10 | 8.9E+09 | 5.9E+08 |
| | 7 | *6/24/2022 12:00 | 1.7E+10 | 2.6E+09 | 5.3E+08 |
| | 8 | 7/3/2022 18:00 | 1.0E+11 | 1.7E+10 | 3.5E+09 |
| | 9 | 7/3/2022 12:00 | 6.0E+10 | 7.3E+09 | 3.7E+08 |
| | 10 | 7/2/2022 18:00 | 8.6E+10 | 1.1E+10 | 8.4E+08 |
| | 11 | 7/2/2022 6:00 | 1.9E+10 | 6.7E+09 | 8.2E+08 |
| | 12 | 4/2/2023 18:00 | 2.4E+10 | 3.9E+09 | 5.6E+08 |
| | 13 | 4/22/2023 0:00 | 2.5E+10 | 1.7E+09 | 4.7E+08 |
| | 14 | 4/30/2023 12:00 | 2.2E+10 | 4.5E+09 | 3.3E+08 |
| | 15 | 6/3/2023 18:00 | 4.2E+10 | 2.8E+09 | 1.6E+10 |
| All Three $T$s | 1 | 11/22/2021 0:00 | 7.7E+08 | 0 | 0 |
| Low | 2 | 11/21/2021 12:00 | 4.0E+08 | 0 | 0 |
| INP | 3 | 11/21/2021 6:00 | 1.4E+09 | 0 | 0 |
| n = 15 | 4 | 11/29/2021 6:00 | 0 | 0 | 0 |
| | 5 | 11/28/2021 18:00 | 0 | 0 | 0 |
| | 6 | 12/29/2021 6:00 | 0 | 0 | 0 |
| | 7 | *1/03/2022 18:00 | 0 | 0 | 0 |
| | 8 | *1/08/2022 18:00 | 0 | 0 | 0 |
| | 9 | 1/14/2022 18:00 | 4.3E+08 | 0 | 0 |
| | 10 | 1/14/2022 12:00 | 9.9E+08 | 0 | 0 |
| | 11 | 1/13/2022 18:00 | 0 | 0 | 0 |
| | 12 | *1/21/2022 06:00 | 0 | 0 | 0 |
| | 13 | *2/02/2022 00:00 | 0 | 0 | 0 |
| | 14 | *2/06/2022 12:00 | 0 | 0 | 0 |
| | 15 | 2/10/2022 12:00 | 0 | 0 | 0 |

Back trajectories are plotted in Figure 9 (for seasonal subsets, see SI Sect. S7). For the 'all three $T$s' case, 15 of 3176 trajectories are considered high INP cases and displayed in Fig. 9b. Some air masses during the high INP period show a westward trajectory from northeastern Alaska. While they appear to pass over the Prudhoe Bay oil field region, distinguishing the influence from that region would require further analysis and attention to resolution beyond the scope here.

Besides Prudhoe Bay, maritime contributions originating from the North Pacific Ocean are a



significant source of high INP trajectories at BRW especially in the summertime (SI Figs. S6 and S7).

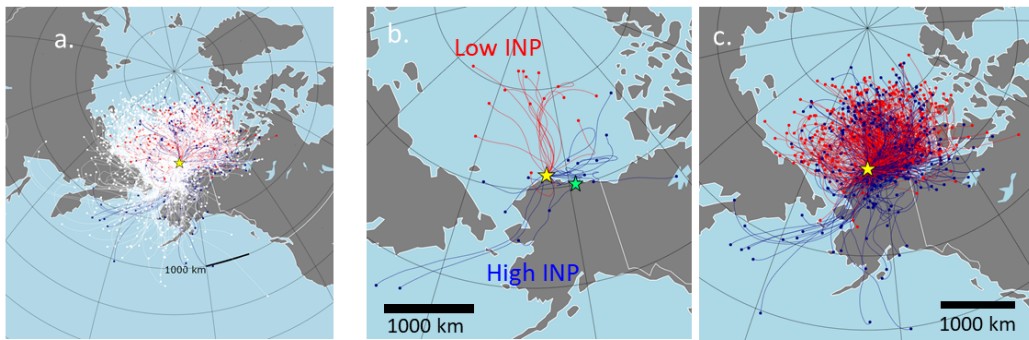

Figure 9. Air mass origins and back trajectories from the inlet height for BRW (yellow star). The Prudhoe Bay location
is indicated by the green star in panel (b). All trajectories for the time period October 2021 − December 2023 are shown in (a). The air mass trajectories during high- and low-INP episodes are shown in blue and red colors. Panel (b) represents the data selected with a low − high threshold of the 25th − 75th percentile based on $n_s(T)$, at all −30, −25, and −20 °C (below or above at 'all three $T$s'). Panel (c) represents the data selected with a low − high threshold of the 25th − 75th percentile based on $n_s(T)$ at any −30, −25, and −20 °C. The details of high- and low-INP episodes in 585 separate panels are shown in SI Figs. S6 and S7. The seasonal breakdowns of the *trajectory data* are shown in SI Figs. S8 and S9.

The high- and low-INP episodes for the any $T$s case based on 72-hour air mass backtrajectories, as displayed in Figure 9c, suggest air mass contributions from North America (particularly the southern Alaska region) and Russian/Siberian Coast are associated with high INP 590 concentrations. A total of 291 and 364 trajectories (out of 3,176) correspond to high- and low-INP events, respectively (Tables 2 and 3). These patterns could suggest that terrestrial sources, potentially influenced by transported biomass burning material in spring and summer, are contributing to the elevated INP levels in BRW. While the exact sources of INPs from high latitudes remain uncertain, previous studies point to biogenic aerosols as a possible source in the 595 Arctic (Inoue et al., 2021; Creamean et al., 2022; Sanchez-Marroquin et al., 2023). In comparison, Moffett et al. (2022) identified the influence of transported biomass burning materials from Russia/Siberia as a key contributor to arctic INP levels, while Irish et al. (2019) reported the presence of INPs in the sea surface microlayer. Similar high INP episodes have been observed in other arctic and sub-arctic regions. For instance, a study in Iceland reported INP concentrations of 600 over 100 L⁻¹ at −26°C (Sanchez-Marroquin et al., 2020), and Southern Alaska showed around 6 L⁻¹ (Barr et al., 2023), reinforcing the importance of dust and other terrestrial sources in these regions.





As seen in Fig. 9c, at BRW low INP episodes coincide with air masses originating from coastal regions of the North American Arctic and contributions from the high Arctic account for > 60 % as compared to other source regions. As discussed in several previous studies (Creamean et al., 2018b; Creamean et al., 2019; DeMott et al., 2016), maritime SSAs are less active as INPs relative to terrestrial dust particles.

## 5. CONCLUSION

Continuous $n_{INP}$ data were measured in the Alaskan Arctic from October 2021 through December 2023. We find a factor of $10 - 1000$ times greater efficiency in the arctic INPs through immersion freezing at sea level during autumn compared to those found previously (Wilbourn et al., 2024) at the mid-latitude ARM sites using the same instrumentation. Specifically, we find relatively low concentrations of aerosol surface area (Fig. 3b) and contrasting high INP concentrations (Fig. 4) at BRW relative to previous observations at the ARM-SGP and ARM-ENA sites. In each of these studies, the same PINE-03 system was deployed for an extended time period. Thus, while the PINE-03 has limitations (for example insensitivity to INPs with freezing temperatures > −16 °C), the relative comparisons among these locations are instructive.

Our analysis of this multi-season INP dataset from BRW offered insight on the variability of INP abundance and revealed seasonality in INP properties. Spring showed profound INP abundance and freezing efficiencies, presumably due to arctic haze events. As previously shown, some arctic haze tracers, such as particulate nss sulfate and nitrate, were found to be higher in spring in our study period than in other seasons. From back trajectory analysis, it is found that air masses of high INP episodes can come from all directions while low INP episodes are strictly from north. More specifically, air masses observed during high INP episodes in spring tended to come from terrestrial regions (Central Alaska). Other than a springtime land contribution, air mass trajectory results also suggest summertime open water and late winter to early spring sea ice regions (the Arctic Ocean and the Pacific Ocean) are potential arctic INP sources. The presence of low pressure over the Aleutian Islands may trigger the transport of warm North Pacific air to northern Alaska (Cox et al., 2019), delivering air masses containing freezing active INPs (local dust). In contrast, low INP episodes identified in this study are dominated by air masses originating from open water in the Arctic Ocean.



Three $n_{\text{INP}}(T)$ datasets were analyzed (i.e., 'all', 'clean', and 'contaminated' data). These datasets are composed of all collected data, screened clean data generated by excluding air sector downwind of nearby settlements and all the data possibly contaminated by operational artifacts, and the segregated flagged data, which is expected to include contamination from Utqiaġvik. Our clean data show very high freezing efficiency of INPs across the measured temperatures as compared to the previous mid-latitude INP measurements made by the same instrument, as well as observed high $n_{\text{INP}}$ above −21 °C throughout the year. The observed high $n_{\text{INP}}$ at high-freezing temperatures occurs in both clean and non-screened datasets (i.e., with or without known local contamination), which suggests the persistent presence of high temperature INPs in Arctic Alaska. Distinct different freezing efficiencies of aerosols observed for the arctic site as compared to the mid-latitude sites indicate the necessity of considering emission source regions yet not to merge whole regions into one because INP data is region dependent.

To contextualize the source of INPs in northern Alaska and the reason for $10 − 1000$ times greater efficiency in the arctic INPs, local and synoptic meteorological influences on INPs must be investigated. In particular, the role of local blowing snow, resuspension of surface materials, and synoptic air mass transport from the warm Pacific Ocean on INPs for selected high- and low-INP episodes can be investigated. Assessing relationships between a regional climate index, known large-scale meteorological patterns influencing northern Alaska, and INP properties will provide an insight on arctic INP properties. Further efforts to correlate INP properties in fall during identified high INP periods with other aerosol and atmospheric parameters will shed additional light on arctic INPs. Such an analysis will be important to comprehensively understand mechanisms and projections of arctic warming beyond the sea ice albedo effect.

Long-term INP datasets, such as that presented here, are lacking in the Arctic but are needed to improve representation of clouds in numerical models. To this end, we developed a parameterization for ice nucleation active surface site density covering −31 to −21 °C. For temperatures higher than −21 °C, INP concentrations were sufficiently low to approach the boundaries of what is detectable given the experimental design, a factor that should be considered for future studies (e.g., examining larger air volume or recreating particle-laden conditions by virtual air mass concentration). This dataset also complements shorter INP datasets previously made in the same region. It will be useful to improve atmospheric models to simulate cloud feedback and determine their impact on the global radiative energy budget. Together with the INP





data, additional aerosol data, such as size-resolved particle chemical composition and mixing state (deployed at BRW in October 2024), would allow us to further understand the implications of this dataset for clouds, precipitation, and regional weather, as well as overall ambient ice nucleation abundance in the NSA region.




## APPENDIX A PREVIOUS STUDIES

Arctic INPs have been reported from the NSA region in several previous studies. The present study reports the first $n_{INP}$ data measured at the BRW site. A summary of 7 studies that report $n_{INP}$ from

or near the NSA is provided in Table A1.

Table A1. A summary of past INP abundance measurements that took place near the BRW monitoring site.

| Study | Measured freezing $T$s (°C) | $n_{INP}$ (L$^{-1}$) | Period | Region | Instrument | Platform |
|---|---|---|---|---|---|---|
| *Present Study* | −16 to −31 | *0.4 to 8.3 **0.6 to 27.0 | Oct. 2021 to December 2023 | NSA | PINE-03 | Ground-Site (BRW) |
| Prenni et al., 2007 (P07) | ≈ −8 to −28 | 0.16 (mean) | Oct. 2004 | NSA | Online CFDC | University of North Dakota's Citation II aircraft |
| Fountain and Ohtake, 1985 (F&O85) | −20 | 0.17 (mean) | Aug. 1978 to Apr. 1979 | NSA | Offline diffusion chamber | Ground-Site (Not specified) |
| Creamean et al., 2018a (C18) | ≈ −5 to −30.5 | ≈ 2.6 x 10$^{-5}$ to 4.4 x 10$^{-2}$ | Mar. to May 2017 | Oliktok Point, NSA | Offline droplet freezing assay | Ground-Site (ARM AMF-3) |
| DeMott et al., 2016 (D16) | ≈ −12 to −20 | ≈ 2.0 x 10$^{-4}$ to 2.0 x 10$^{-2}$ | Summer 2012 | Central Bering Sea | Online CFDC | Research Vessel Araon |
| Sanchez-Marroquin et al., 2023 (S-M23) | ≈ −14 to −30 | ≲ 40 | Mar. 2018 | NSA coast to Yukon, Canada | Offline droplet freezing assay | UK's BAe-146 FAAM Aircraft |
| Rogers et al., 2001 (R01) | −10 to −30 | ≲ 57 | May 1998 | Offshore NSA | Online CFDC | NCAR C-130 Aircraft |
| Inoue et al., 2021 (I21) | ≈ −7.5 to −29.5 | ≈ 5.0 x 10$^{-4}$ to 10$^{2}$ | Nov. 10–21, 2018 | Chukchi Sea NSA | Offline droplet freezing assay | Research Vessel Mirai |

*clean median; **clean average; the data screening protocol is described in Sect. 2E.

## APPENDIX B BACKTRAJECTORY ANALYSIS

Trajectories were based on the Global Data Assimilation System (GDAS) and calculated using the

Hybrid Single-Particle Lagrangian Integrated Trajectory (HYSPLIT) model (Rolph et al., 2017; Stein et al., 2015) to compute archive trajectories every 6 hours during the sampling period. Each 72-hour backward trajectory was calculated at the sampling inlet height (~ 12 m AGL). Our analysis protocols follow those of Wilbourn et al. (2024). Back trajectory origins were classified into broad regional categories, including the major oceans and continents, as described in SI Sect.

12 of Wilbourn et al. (2024).

Source points are assigned to the final back trajectory locations at 72-hr. Besides land and ocean, we also determine if the source was over an area covered in sea ice. The sum of rainfall is calculated at each height, and if the rainfall amount exceeds 7mm, then the back trajectory point



before exceeding 7 mm rainfall is used as the source point. If it does not exceed 7mm, the 72-hour

point is used. More information on the analysis of air mass travel times over different surface types

(land, open water, and ice) and the impact of precipitation  (presuming > 7 mm cumulative rainfall

can wash out aerosols in air mass by wet scavenging) can also be found in Wilbourn et al. (2024)

and Gong et al. (2020).

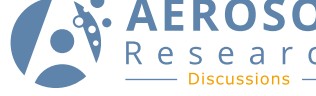

ACKNOWLEDGMENTS:

This research was supported by the US Department of Energy, Office of Science, Office of Biological and Environmental Research (grant no. DE-SC-0018979). The authors gratefully acknowledge the NOAA Air Resources Laboratory (ARL) and Environmental Research Division's Data Access Program (ERDDAP) for the provision of the HYSPLIT transport and dispersion model and/or READY website (https://www.ready.noaa.gov) and the NOAA/PMEL ion

chromatography data used in this publication. A useful discussion regarding the ERDDAP data with NOAA's Lucia Upchurch and Patricia Quinn is also acknowledged. The authors acknowledge Valerie Sparks with Sandia National Labs for her onsite and administrative contributions to the ExINP-NSA campaign. Naruki Hiranuma, Aidan D. Pantoya, and Stephanie R. Simonsen thank Elise Wilbourn, Andebo A. Waza, Jacob Hurst, and Oyshrojo Talukder for their contribution to

the PINE-03 operation, maintenance, and error analysis. The authors also acknowledge Amy Solomon, Gourihar Kulkarni, and Matthew Shupe for useful discussion regarding the data implications. The scattering and aerosol size distribution data from El Arenosillo used to calculate Q was downloaded from the EBAS data archive. Gijs de Boer was supported by the US Department of Energy Atmospheric System Research (ASR) program under cooperative

agreement DE-SC0013306, as well as by the NOAA Physical Sciences Laboratory. Christopher J. Cox received support from NOAA's Global Ocean Monitoring and Observing Program through the Arctic Research Program (FundRef https://doi.org/10.13039/100018302) and the NOAA Physical Sciences Laboratory. Elisabeth Andrews was supported partly by the NOAA cooperative agreement with CIRES (NA17OAR4320101) and partly by the Atmospheric Radiation

Measurement (ARM) user facility, a US Department of Energy (DOE) Office of Science user facility managed by the Biological and Environmental Research program.

DATA AVAILABILITY:

The dataset and codes created for the study will be available at https://doi.org/10.6084/m9.figshare.26615752.v3 (Pantoya and Hiranuma, 2024). Local ambient

conditions, such as wind speed, wind direction, temperature, and relative humidity, were obtained from NOAA-GML (https://gml.noaa.gov/aftp/data/meteorology/in-situ/brw/, last access: 10 June 2024). Visibility and time-averaged cumulative precipitation observations were not available at BRW for our study period and reported from Wiley Post-Will Rogers Memorial Airport (71.285° N, 156.769° W) (automated monitoring station; https://mesowest.utah.edu/cgi-

bin/droman/download_api2.cgi?stn=PABR&hour1=03&min1=35&timetype=LOCAL&unit=0&



graph=0, last access: 10 June 2024). Ion analysis data used for estimating ambient mass concentrations of nss $SO_4^=$ and aerosol $NO_3^-$ are available from NOAA-PMEL's website (https://saga.pmel.noaa.gov/data/stations/, last access: 18 July 2024). Aerosol data from BRW are downloaded from NOAA-GML (https://gml.noaa.gov/aftp/aerosol/brw/, last access: 10 June

725 2024).

SUPPLEMENT:
The supplement related to this article is available online at: https://www.aerosol-research.net/

AUTHORS CONTRIBUTION:
NH designed the concept of this collaborative research. The methodology was developed by ADP,

SRS, and NH. The onsite and remote measurements at the BRW site were conducted by BDT, RB, and ADP. The formal data analyses were led by ADP with the data processing contributions of SS, EA, CJC, and GdB. ADP, SRS, and NH led the writing of the manuscript with the support of all authors.

COMPETING INTERESTS:

The authors declare no conflict of interest.



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
