# Peer review of "Multi-seasonal measurements of the ground-level atmospheric ice-nucleating particle abundance on the North Slope of Alaska"

_Aerosol Research, 2025_

## Author Comment (AC1)

RC1: Review in black text, responses in blue. Additional table and figure prepared for the authors response document may contain other colors.

RC: Overall Evaluation: I have reviewed the manuscript "Multi-seasonal measurements of the ground-level atmospheric ice-nucleating particle abundance on the North Slope of Alaska". This manuscript presents a comprehensive analysis of atmospheric ice-nucleating particles (INPs) based on long-term ground-based measurements in the Alaskan Arctic. The study is significant as it provides one of the first multi-seasonal datasets of INP abundance in this region. The authors utilize a portable ice nucleation experiment chamber (PINE-03) to collect high-resolution data over a two-year period, and analyze aerosol and meteorological data to assess the correlation between ambient $n$INP, air mass origin region, and meteorological variability. The manuscript is well-written, with clear language and logical organization. The introduction effectively sets up the research question, and the conclusion summarizes key findings concisely. While the study is well-structured and the results are relevant to the field of aerosol-cloud interactions, several aspects require further clarification and improvement. I believe that the conclusions of the manuscript are likely valid, and the manuscript is publishable subject to minor revisions. I have some specific concerns, listed below.

AR: We appreciate this thorough and thoughtful review. We believe that the quality of this paper have improved with the changes made to the current version of the manuscript. Below, we provide our point-by-point responses.

RC: Specific comments: The manuscript presents substantial new data on Arctic INPs, which fills a critical gap in long-term observations, especially using high resolution instruments. However, while the manuscript provides a novel dataset, the interpretation of the sources of INPs could be strengthened. The manuscript concludes high INP concentration in spring possibly related to arctic haze. It could be improved by incorporating a arctic haze event with high resolution INP measurement and aerosol , meteorological data.

AR: **Aerosol Data:** The time resolution of ambient mass concentration measurements of major arctic haze tracers, such as non-sea salt (nss) $SO_4^=$ and aerosol $NO_3^-$, from BRW was equivalent to or longer than 24 hours. Further, the sampling interval was not consistent, preventing high-resolution arctic haze tracer – INP correlation analysis. Other complementary aerosol composition measurements (e.g., aerosol mass spectrometer) were not available during our study period. While we observed that both INP and arctic haze tracer concentrations are relatively high in spring as compared to other seasons (Figs. 3 and 5), the correlation between these two aerosol populations based on their seasonal averages is not significant ($r^2 < 0.2$). Thus, the INP-arctic haze relation is not conclusive. The authors intended to keep a softened qualitative tone regarding the relation in the current manuscript. We wish to keep the relevant discussions.

L529: To mitigate the reviewer's concern, we now added, "…while further quantitative analysis with high-time-resolution data is necessary." We agree that highly time-resolved analysis of INP with appropriate, complementary aerosol composition would be necessary in the future.

**Meteorological Data:** Table AR1 compares all trajectories (N = 3176), all three $T$s back trajectories (N = 30), and any $T$s air trajectories (N = 654) during high- and low-INP episodes. We made two subsets of high- and low-INP episodes; one where all three temperatures (−20, −25, and −30 °C) had to exceed the percentile thresholds ('all three $T$s') and another with the same thresholds but where the sample qualified if 'any' of the three examined temperatures met the threshold. For the former case, we identified 15 high INP episodes and 15 low INP episodes. For the latter case, we identified 291 data points as being in a high INP period and 364 as being in a low INP period (SI Table S3). According to our HYSPLIT back trajectory analysis (see Appendix B and SI S7), **the time fraction of air mass over land, especially in North America, accounted for more than 15%. Spring maxima of land and North American terrestrial contributions, accounting for 27% and 24% of air masses, imply high $n_{INP}$ might be associated with land origins.** Further, for high-INP periods, North American land origins contribute

substantially (>20%), whereas this contribution is minimal during low-INP periods (<7%). Eurasian land origins are minor and likely insignificant for INP delivery to the NSA. Open water air masses are more frequent in low-INP episodes (~33 – 40%) as compared to high-INP episodes (~13 – 20%). At BRW, the Arctic Ocean north of 66° N are the main air mass origins, but contribute more to low-INP periods (91 – 100%) than to high-INP episodes (≈48 – 71%).

Table AR1. Percentage of air mass origin region, as well as air mass time fractions over open water, land, or ice, determined from 72-hour HYSPLIT back trajectories (back trajectories may be younger than 72 hours if rainfall exceeds 7mm). For each dataset (i.e., all three $T$s and any $T$s), each column represents air mass properties for all trajectories, high INP periods, and low INP periods. The numbers in the parenthesis represent seasonal minimum – maximum color coded as follows: fall – orange, winter – blue, spring – green, and summer − red. The season-segregated tables are available in SI Sect. S7.

| ORIGIN | All Trajectories All $N = 3176$ (720−864) | All Three $T$s High INP period $n = 15$ (0−10) | All Three $T$s Low INP period $n = 15$ (0−10) | Any $T$s High INP period $n = 291$ (43−97) | Any $T$s Low INP period $n = 364$ (25−159) |
|---|---|---|---|---|---|
| Arctic Ocean North of 66°N Latitude | 70.4 (61.8−76.3) | 48.0 (0−64.0) | 100 | 70.5 (45.4−77.6) | 91.2 (69.1−96.0) |
| Arctic Ocean South of 66°N Latitude | 5.9 (3.1−8.6) | 12.0 (0−16.0) | 0 | 7.8 (3.7−9.5) | 1.0 (0−2.9) |
| North America | 15.6 (5.2−23.6) | 26.7 (20.0−40.0) | 0 | 21.4 (12.4−39.7) | 6.9 (2.9−24.0) |
| Norwegian Sea | 0 | 0 | 0 | 0 | 0 |
| Pacific Ocean | 4.2 (1.5−8.9) | 13.3 (0−20.0) | 0 | 6.6 (2.3−11.3) | 0.3 (0−4.0) |
| Eurasia | 3.9 (1.4−5.0) | 0 | 0 | 0.3 (0−2.3) | 0.6 (0−1.3) |
| Western Africa | 0 | 0 | 0 | 0 | 0 |
| Land | 19.4 (10.0−27.1) | 26.1 (20.8−36.7) | 5.0 (0−7.5) | 21.8 (15.6−37.9) | 8.0 (5.2−18.7) |
| Open Water | 28.6 (19.8−38.4) | 12.8 (8.3−15.0) | 40.0 (36.7−41.7) | 19.7 (12.8−38.4) | 32.9 (18.7−44.9) |
| Ice | 51.9 (39.4−61.6) | 61.1 (55.0−64.2) | 55.0 (50.8−63.3) | 58.6 (34.9−70.4) | 59.1 (49.9−67.4) |

RC: Line 110-112. In the first paragraph, I recommend providing additional details about the INP dataset, particularly addressing the significant data gaps observed during certain months, as mentioned in lines 386–387. This would help clarify the dataset's continuity and any potential implications for the study's conclusions.

AR: This is a valid suggestion. L117- now reads "We note that any data gaps pertain to the PINE-03 system maintenance, as required every 3 − 4 months (see Wilbourn et al., 2024; SI Sect. S5). The maintenance was also conducted immediately after we observed and flagged the PINE-03 operational issues. The most common problems include an OPC malfunction, diaphragm pump filter replacement, or LabView data acquisition console disconnection. During ExINP-NSA, we rarely observed such issues (41 out of 1506 operations, 2.7%), and PINE-03 ran reliably with scheduled maintenance periods. Operational flagging was assessed every cycle during measurements".

RC: Line 134. In addition to discussing the time resolution, it would be beneficial to include information about the size range of aerosols that PINE-03 measures. This would provide a more comprehensive understanding of the instrument's capabilities and its relevance to INP measurements.

AR: This is also a valid suggestion. Both INP and total aerosol abundances were measured through the inset stack inlet. The upper size of measurable $n_{ear}$ and $n_{INP}$ is limited by $D_{50}$ of particles passing through

the stack inlet, which is <3 μm (Sect. S1). The $n_{ear}$ measurement is based on a CPC, which has a measurable size range from ≈0.01 to ≈3.0 μm. The lower bound of measurable particle size is limited by a diffusion loss of particles through the inlet and should be consistent for both INP and total aerosols. Note that, while we cannot define the lower bound of measurable INP size, small aerosols provide small surfaces, which do not contain as many active sites as on larger particles (unless it is known ice nucleation active biological particles). The lower size limit of homogeneous freezing can be as small as the size of a water cluster of 100-300 water molecules (<10 nm), but it occurs below -35 °C, which is outside of our measured freezing temperature ranges. This discussion is now added to SI S1 as it is relevant to the measured particle loss discussion.

RC: Line 275-285. As described in the manuscript and shown in Figure 1, the winter temperature in the study region is consistently below -20°C, with a recorded minimum of -37.2°C. Additionally, the relative humidity ranges from 60% to 80%, indicating that the dew point temperature is even lower. Given these conditions, I have concerns regarding the measurement of INPs that are activated at relatively higher temperatures (-16°C to -31°C). Could the authors clarify how the PINE-03 system measures INP activation at these temperatures under such ambient conditions? Specifically, can PINE-03 create chamber conditions where the INP. activation temperature is higher than the ambient air temperature? Further explanation of this aspect would enhance the understanding of the instrument's capabilities and potential limitations.

AR: The authors assessed the $n_{INP}$ data for two conditions; (1) dew point temperature in the chamber vessel (DPT) higher than PINE-03 freezing temperature (Freezing T) and (2) DPT lower than Freezing T at three selected temperatures (-20, -25, and -30 °C) to be consistent with our high vs. low INP episode analysis. We plotted the comparison using (a) All $n_{INP}$ data, and (b) Clean $n_{INP}$ data, and (c) Clean Winter $n_{INP}$ data. As seen in the figure below, we observed a difference in $n_{INP}$ at each examined temperature for the two conditions. We see higher $n_{INP}$ for the dataset of DPT > Freezing T. However the difference is within the standard deviation for all data assessed and thereby not conclusive.

[Figure]

The authors note that Wilbourn et al. (2024)* state that PINE-03 is capable of measuring but not distinguishing between both immersion-mode and deposition-mode freezing events. Möhler et al. (2021)** reported that PINE-03 is capable of detecting pore condensation freezing and deposition freezing processes. These freezing modes may be seen when the chamber is supersaturated with respect to ice yet under a water-subsaturated condition. Thus, the much larger discrepancy in nINP at BRW may be due to low vs. high DPT. For instance, the immersion mode freezing cloud has been missed when DPT < Freezing T. However, as all previous PINE work at multiple sites has counted total INPs, we will report the same for this manuscript. This remains an area of uncertainty that could be examined by future researchers. We note that this important limitation is addressed in L538-. "PINE-03 is designed to utilize ambient moisture to saturate the chamber during expansion cooling and for maintaining the chamber dew point temperature above freezing temperature. Dry winter conditions often lowered dew point and hindered INP measurements." We believe we carefully phrased our measurable freezing definition in the current manuscript.

*Wilbourn, E. K. et al.: Measurement report: A comparison of ground-level ice-nucleating-particle abundance and aerosol properties during autumn at contrasting marine and terrestrial locations, Atmos. Chem. Phys., 24, 5433–5456, https://doi.org/10.5194/acp-24-5433-2024, 2024.
** Möhler, O. et al.: The Portable Ice Nucleation Experiment (PINE): a new online instrument for laboratory studies and automated long-term field observations of ice-nucleating particles, Atmos. Meas. Tech., 14, 1143–1166, https://doi.org/10.5194/amt-14-1143-2021, 2021.

RC: Line 286. What is the significance of averaging visibility in this context? For example, the average of a 10 km visibility and a 100 m visibility would be around 5 km, but this may not accurately reflect the actual atmospheric conditions. For INP analysis, it may be more relevant to focus on the statistics of low-visibility events, as these could be associated with fog, blowing snow, or dust events. Could the authors provide further clarification on this aspect?

AR: To mitigate the reviewer's concern, we removed the visibility and wind properties data points from Fig. 1. Our thoughts regarding the relevancy and importance of localized events to INP are consistent with the reviewer's comment. The point we wanted to make here was the potential importance of localized events listed in the manuscript. Visibility and wind direction data fluctuated throughout the study period, which could have induced localized aerosol emissions. Several authors of this manuscript are planning to submit another paper focusing on the impact of local blowing snow and other larger-scale meteorological events in the NSA region (Chen et al., 2022*) on ambient INP abundance at the ground level in the future (too much to be included in a single manuscript).

*Chen, Q. et al.: Atmospheric particle abundance and sea salt aerosol observations in the springtime Arctic: a focus on blowing snow and leads, Atmos. Chem. Phys., 22, 15263–15285, https://doi.org/10.5194/acp-22-15263-2022, 2022.

[Figure]

Figure 1. The time series of the 6-hour average temperature and relative humidity (a). Panel (b) displays the 6-hour average precipitation and monthly cumulative precipitation amounts. Dashed lines in each panel are mean seasonal values of individual measurements and the green shaded area represents spring. Precipitation data from mid-August 2023 was not available, which is indicated by the grey shaded area. The relative humidity data from late August to early December 2022 is also missing.

RC: Figure 1. In Figure 1, are the wind direction and wind speed also averaged over six hours? Since wind direction and wind speed are vector quantities, they may not be directly suitable for simple averaging. Could the authors clarify how these values were processed?

AR: We calculated vector averaged wind speed. Nevertheless, we decided to delete Panel B as Fig. 2 shows wind properties. All associated numbers in this manuscript are based on vector average numbers.

[Figure]

RC: The x-axis in Figure 4 is too dense; I suggest adjusting it for better readability. Additionally, I recommend removing the black error bars, as they affect the visual aesthetics of the figure without significantly enhancing the clarity of the authors' intended message.

AR: Done (see the next page).

[Figure]

Figure 4. INP concentrations ($n_{INP}(T)$) measured at BRW. The 'all' dataset collected throughout the campaign is shown in (a). The segregated datasets collected during the 'clean' periods (clean data) and 'contaminated' periods (presumably contaminated data) are shown in (b) and (c). Each point represents a 6-hour time-averaged concentration. The color scale indicates the measured freezing temperature. Individual data points are temperature binned for 1 °C. The campaign mean and median $n_{INP}(-25°C)$ are shown with dark blue and cyan lines.

RC: Line 475-484. I suggest moving the explanation of the instrument's resolution to Section 2B, which discusses INP measurement data. Additionally, all data presented in the figures should be carefully selected and evaluated to support the study's conclusions.

AR: Former L475-476 is moved to L188-189. The rest remains as is as they are relevant in this particular section.

RC: Line 487. Regarding the comparison of Ns with previous studies, could the observed differences be attributed to variations in the calculation methods used for Ns?

AR: No. The calculation method is consistent. It's mainly due to the difference in abundance of surface area.

RC: Additionally, are the surface area results obtained using the method in this study consistent with those measured by aerosol spectrometers (APS and SMPS)? Could the authors provide further clarification on this matter?

AR: The effective aerosol scattering efficiency ($Q_{eff}$) we obtained was consistent with the APS (coarse) and SMPS (fine) comparisons with the nephelometer as discussed in SI Sect. S3. To mitigate the reviewer's concern, we added the following sentence in the Table S2 caption – "The $Q_{eff}$ value in the table is from the nephelometer and APS comparison."